# Malicious Package Detection using Metadata Information

## ABSTRACT

Protecting software supply chains from malicious packages is paramount in the evolving landscape of software development. Attacks on the software supply chain involve attackers injecting harmful software into commonly used packages or libraries in a software repository. For instance, JavaScript uses Node Package Manager (NPM), and Python uses Python Package Index (PyPi) as their respective package repositories. In the past, NPM has had vulnerabilities such as the event-stream incident, where a malicious package was introduced into a popular NPM package, potentially impacting a wide range of projects. As the integration of third-party packages becomes increasingly ubiquitous in modern software development, accelerating the creation and deployment of applications, the need for a robust detection mechanism has become critical. On the other hand, due to the sheer volume of new packages being released daily, the task of identifying malicious packages presents a significant challenge. To address this issue, in this paper, we introduce a metadata-based malicious package detection model, *MeMPtec*. This model extracts a set of features from package metadata information. These extracted features are classified as either easy-to-manipulate (ETM) or difficult-to-manipulate (DTM) features based on monotonicity and restricted control properties. By utilising these metadata features, not only do we improve the effectiveness of detecting malicious packages, but also we demonstrate its resistance to adversarial attacks in comparison with existing state-of-the-art. Our experiments indicate a significant reduction in both false positives (up to 97.56%) and false negatives (up to 91.86%).

## CCS CONCEPTS

• **Do Not Use This Code** → **Generate the Correct Terms for Your Paper**; *Generate the Correct Terms for Your Paper*; Generate the Correct Terms for Your Paper; Generate the Correct Terms for Your Paper.

## KEYWORDS

NPM Metadata, Malicious Detection, Feature Extractions, Adversarial Attacks, Software Supply Chain

**ACM Reference Format:**
Anonymous Author(s). 2018. Malicious Package Detection using Metadata Information. In *Proceedings of Make sure to enter the correct conference title from your rights confirmation email (Conference acronym 'XX)*. ACM, New York, NY, USA, 11 pages. https://doi.org/XXXXXXX.XXXXXXX

## 1 INTRODUCTION

Nowadays, Free and Open-Source Software (FOSS) has become part and parcel of the software supply chain. For example, the Open Source Security and Risk Analysis (OSSRA) report in 2020 shows that as much as 97% of codebases contain open-source code [23] and the proportion of enterprise codebases that are open-source increased from 85% [21] to 97% [25]. Thus, modern software developers thrive through the opportunistic reuse of software components that save enormous amounts of time and money. The node package manager (NPM) offers a vast collection of free and reusable code packages to support JavaScript developers. Since its inception in 2010, NPM has grown steadily and offers over 3.3 million packages as of September 2023 [14]. The extensive library of packages provided by NPM is a valuable resource for developers worldwide and is expected to continue growing. Different from JavaScript, Python uses Python Package Index (PyPi) as their package repositories. Both NPM and PyPi have faced security vulnerabilities in the past, such as the event-stream incident, where a malicious package was introduced into a popular NPM package, potentially impacting a wide range of projects. Similarly, PyPi has experienced concerns with typo-squatted packages that appear similar to common libraries but contain malicious code, posing a risk of inadvertent installation by developers. Therefore, detecting malicious packages is essential to protect software supply chains.

Metadata associated with package repositories plays a crucial role in the software development lifecycle. Such metadata includes information about the creator, update history, frequency of updates, and authorship, among other details. This information can be indicative of maliciousness within packages, for example, a package that has unknown authors is likely to be malicious [9]. However, such heuristics are not sufficient as attackers can intentionally compromise metadata information to bypass detection models. Thus, extracting a set of features that are both predictive yet resistant to adversaries seeking to game the model is critical. There are several advantages of using metadata feature selection to detect malicious packages. First, it can help identify malicious packages quickly without requiring extensive manual review. Second, it is efficient because it can detect malicious packages faster than thorough code analysis. Third, metadata analysis can be used to gain insights into behavioural patterns of malicious packages in large datasets. Lastly, the incorporation of metadata features increases model resilience against adversarial attacks, offering a more robust defense mechanism compared to existing state-of-the-art methods.

There are some existing research works that utilise metadata information. For example, by using metadata information, Zahan et al. [31] introduced a model for measuring NPM supply chain weak link signals to prevent future supply chain attacks. However, they do not consider the challenge of adversarial attacks. The main motivation of this research is to propose a model to detect malicious packages in the NPM repository to protect software developers, organizations, and end-users from security breaches that can result from downloading and using packages containing malicious code.

As the NPM repository is widely used to store and distribute open-source packages, it is an attractive target for attackers looking to compromise the security of a large number of systems. By detecting malicious packages in the repository, organizations can ensure that their software development processes are not disrupted and that security threats do not compromise their systems. Detecting malicious packages also helps maintain the trust and integrity of open-source package repositories, which are essential for the long-term success and growth of the software development community.

In this paper, we address the following research questions.

- **RQ1:** How can metadata information be effectively leveraged to accurately identify malicious packages within repositories?
- **RQ2:** How can the robustness of metadata-based detection models be enhanced against adversarial attacks?

To address these research questions, a metadata based malicious package detection model is developed. The main contributions of our research work are as follows:

- We propose an advanced metadata based malicious package detection (*MeMPtec*) model leveraging new metadata features and machine learning algorithm.
- We introduce a new metadata feature extraction technique which partitions features into easy-to-manipulate and difficult-to-manipulate.
- We investigate stakeholder based adversarial attacks and propose adversarial attack resistant features based on monotonicity and restricted control properties.
- We conduct extensive experiments[1] that show our proposed *MeMPtec* outperforms the existing feature selection strategies from the state-of-the-art in terms of precision, recall, F1-score, accuracy and RMSE. It reduces false positives on average by 93.44% and 97.5% in balanced data and imbalanced data, respectively, and reduces false negatives on average by 91.86% and 80.42% in balanced and imbalanced data, respectively.

## 2 EXISTING WORKS

In this research work, we have focused on attack detection utilising metadata in NPM ecosystems. Works on attack detection and remediation include the followings. Liu et al. [10] introduced a knowledge graph-driven approach for dependency resolution that constructs a comprehensive dependency-vulnerability knowledge graph and improved vulnerability remediation method for NPM packages. Zhou et al. [34] enriched the representation of Syslog by incorporating contextual information from log events and their associated metadata to detect anomalies behaviour in log files. Zaccarelli et al. [29] employed machine learning techniques to identify amplitude anomalies within any seismic waveform segment metadata, whereas the segment's content (such as distinguishing between earthquakes and noise) was not considered. Anomaly detection on signal detection metadata by utilising long and short-term memory recurrent neural networks in the generative adversarial network has been introduced in [3]. Mutmbak et al. [12] developed a heterogeneous traffic classifier to classify anomalies and normal

behaviour in network metadata. Pfretzschner et al. [17] introduced a heuristic-based and static analysis to detect whether a Node.js is malicious or not. Garrett et al. [5] proposed an anomaly detection model to identify suspicious updates based on security-relevant features in the context of Node.js/NPM ecosystem. Taylor et al. [24] developed a tool named *TypoGard* that identifies and reports potential typosquatting packages based on lexical similarities between names and their popularities.

Some efforts have been devoted in the literature to detect malicious attacks using metadata features. For example, Abdellatif et al. [1] utilised metadata information for the packages' rank calculation simplification. Zimmermann et al. [36] have demonstrated a connection between the number of package maintainers and the potential for introducing malicious code. Scalco et al. [19] conducted a study to assess the effectiveness and efficiency of identifying injected code within malicious NPM artifacts. Sejfia et al. [20] presented automated malicious package finder for detecting malicious packages on NPM repository by leveraging package reproducibility checks from the source. Vu et al. [26] applied metadata to identify packages' reliability and actual sources. Ohm et al. [15] investigated limited metadata information (e.g., package information, dependencies and scripts) to detect malicious software packages using supervised machine learning classifiers. However, these approaches do not address the issue of adversarial attacks, and as demonstrated by our experiments (*c.f.* Section 6.3), the features proposed in the literature are prone to adversarial manipulation.

Other works related to software security, but not metadata based include [22, 28, 30, 32, 35]. Zahon et al. [30] compared the security practices of NPM and PyPI ecosystems on GitHub using Scorecard tools that identifies 13 compatible security metrics and 9 metrics for package security. Sun et al. [22] introduced *CoProtector*, a tool designed to safeguard open-source GitHub repositories from unauthorized use during training. Wi et al. [28] proposed a scalable system that detects web vulnerabilities, such as bugs resulting from improper sanitization, by employing optimization techniques to tackle the subgraph isomorphism problem. Zhang et al. [32] developed *GERAI*, which uses a differential private graph convolutional network to protect users' sensitive data from attribute inference attacks. Zhu et al. [35] built a system that uses various information types to detect spam reviews.

### 2.1 Differences with Previous Works

Our proposed malicious detection based on metadata information differs from state-of-the-art malicious detection techniques in various aspects. Firstly, we categorise the different sets of features that can be derived from metadata information, whereas existing methods considering metadata information do not make a distinction between the types of metadata features that can be extracted. Secondly, we consider the problem of adversarial attacks and introduce the concept of difficult-to-manipulate (*DTM*) features that reduce the risk of adversarial attacks. Table 1 highlights some key differences between features derived from our approach versus those proposed in the literature. In the foregoing, we use the term *Existing_tec wrt* metadata features to refer collectively to the sets of features proposed in the literature for malicious package detection.

---

[1]All code and datasets associated with this paper will be made available upon paper acceptance.

**Table 1: Comparison between the types of metadata features considered: literature vs. our approach.**

| Research Work | Descriptive | Stakeholders | Dependencies | Repository | Temporal | Package Interaction |
|---|---|---|---|---|---|---|
| Zimmermann et al. [36] | | ✓ | | | | |
| Abdellatif et al. [1] | | ✓ | ✓ | ✓ | | |
| Zahan et al. [31] | ✓ | ✓ | ✓ | | | |
| Ohm et al. [15] | ✓ | | ✓ | | | |
| Vu et al. [26] | ✓ | | | | | |
| **Existing_tec** [1, 15, 26, 31, 36] | ✓ | ✓ | ✓ | ✓ | | |
| **MeMPtec_E** | ✓ | ✓ | ✓ | ✓ | | |
| **MeMPtec_D** | | | | | ✓ | ✓ |
| **MeMPtec** (Proposed) | ✓ | ✓ | ✓ | ✓ | ✓ | ✓ |

## 3 PRELIMINARIES AND PROBLEM STATEMENT

In this section, we present the preliminary notations and definitions.

Let $P = \{p_1, \cdots, p_n\}$ be set of packages. A package often involves several participants, namely *author, maintainer, contributor, publisher*. We refer to these collectively as stakeholders denoted by $S_i = \{s_j\}_i$ such that $s_{j_i}$ is a stakeholder of type $j$ involved in package $p_i$.

DEFINITION 1 (PACKAGE METADATA INFORMATION (PMI)). *Given a package $p_i \in P$, the package metadata information denoted $\mathcal{I}_i = \{\langle k, v \rangle\}$ is the set of key-value pairs of all metadata information associated with $p_i$.*

In this work, without loss of generality, we adopt the NPM package repository as an exemplar due to its popularity in web applications and various cross platforms [20]. Table 2 shows our considered NPM package metadata information.

**Table 2: Package Metadata Information.**

package_name, version, description, readme, scripts, distribution_tag, authors, contributors, maintainers, publishers, licenses, dependencies, development_dependencies, created_time, modified_time, published_time, NPM_link, homepage_link, GitHub_link, bugs_link, issues_link, keywords, tags and topics.

DEFINITION 2 (PROBLEM DEFINITION). *Given a package $p_i \in P$ and its package metadata information $\mathcal{I}_i = \{\langle k, v \rangle\}$, the goal is to develop a malicious package detector $\mathcal{M}$ as follows:*

$$\mathcal{M}(p_i, \mathcal{I}_i) = \begin{cases} 1, & \text{if } p_i \text{ is malicious,} \\ 0, & \text{otherwise} \end{cases}$$

There are three key challenges to address in the problem definition above. Firstly, the PMI of each package may contain several pieces of information, some of which may be irrelevant to the detection task, and it may also have inconsistent representation across different packages (**Challenge 1**). For example, packages may contain copyright and browser dependencies that are often not relevant for detecting malicious packages. Secondly, metadata information may be prone to manipulation by an adversary who wishes to evade detection by a detection model $\mathcal{M}$ (**Challenge 2**). Thirdly, for any detection model $\mathcal{M}$ to be practical, it needs to achieve high true positive rates with low false positive rates (**Challenge 3**).

To address the above challenges, we propose a novel solution called Metadata based Malicious Package Detection (MeMPtec). MeMPtec relies on a feature engineering approach to address the aforementioned challenges. This is detailed in the following sections.

## 4 CATEGORISATION OF PACKAGE METADATA INFORMATION

Each piece of information contained in PMI represents a different type of information. In this section, we categorise each PMI in order to understand its relevance for malicious package detection. This is important because not all information in metadata packages is crucial for malicious package detection (**Challenge 1**). We consider the following categories.

- **Descriptive Information**: This includes information that describes the resource, such as package title, versions, description, readme, scripts and distribution tag.
- **Stakeholder Information**: It provides information about the individuals or organizations involved in developing, maintaining and distributing a package. Some stakeholder information includes authors, contributors, maintainers, publishers and licenses.
- **Dependency Information**: Dependency information provides details about the external packages or modules that a particular package depends on. These include dependencies and development dependencies.
- **Provenance Information**: It provides information about when various events related to the package occurred. This information can be useful for tracking the package's history and understanding how it has evolved over time. For example, package created, modified and published time information.
- **Repository Information**: It provides information about the location of the source code repository for a package, such as the NPM link, homepage link and GitHub link.
- **Context Information**: Context information provides additional information based on their functionality and purpose. For example, keywords, tags and topics.

Table 3 presents the categories of package metadata information.

## 5 FEATURE EXTRACTION AND SELECTION

It is necessary to extract features from the PMI for each package to generate a consistent set of features for all packages. For example, let $\langle package\_name, generator@1 \rangle \in \mathcal{I}$ be a toy example of a key-value pair in the PMI $\mathcal{I}$. The package name is *generator@1*, but we can derive features from this package name, such as whether or not it contains a special character or the length of the package name. These features are of particular relevance in the context of detecting malicious packages since package names play a crucial role in identifying combosquatting and typosquatting [27]. Thus,

**Table 3: Package metadata information and their corresponding categories.**

| Category | Information Name |
| --- | --- |
| Descriptive | Package name, Versions, Description, Readme and Scripts |
| Stakeholder | Authors, Contributors, Maintainers, Collaborators and Publishers |
| Dependency | Package Dependencies and Development Dependencies. |
| Provenance | Package Created, Modified and Published time. |
| Repository | NPM Link, Homepage Link, GitHub Link, Bugs Link and Issues Link. |
| Context | Keywords, Distribution Tag and Licenses. |

feature extraction in our context is a one-to-many mapping between PMI and a set of features that is formally defined as follows:

DEFINITION 3 (FEATURE EXTRACTOR $\mathcal{F}$). *Given a package $p_i$ and its associated PMI containing $\ell$ key-value pairs, $I_i = \{\langle k, v \rangle_1 \cdots \langle k, v \rangle_\ell\}$, a feature extractor denoted $\mathcal{F}$ is a multivalued function which maps each PMI $\langle k, v \rangle_j$ unto one or more features in the set $X$:*

$$\mathcal{F} : I_i \mapsto X.$$

As noted earlier, one of the challenges to be addressed while developing malicious package detector $\mathcal{M}$ is its ability to resist adversarial attacks (**Challenge 2**). We define an adversary as follows:

DEFINITION 4 (ADVERSARY). *An adversary $\mathcal{A}$ is any stakeholder in $S_i$ of package $p_i$ who has the authority to modify the metadata information $I_i$ of package $p_i$, and attempt to do so to evade detection by a model.*

The definition above represents the scenario where a stakeholder of a malicious package may alter the metadata information to evade detection by a model. We make the following assumption and then present two important properties relevant to the feature $x \in X$.

ASSUMPTION 1. *Given a repository environment such as the NPM package repository, we assume that all security protocols are intact, and users follow the protocols to engage with the repository environment i.e. there is no subversion of the system by an adversary.*

This assumption is pivotal to our approach and indeed to any metadata-based malicious package detection technique, including [5, 6, 9]. If this assumption does not hold, then it renders metadata information useless for any purpose. At the same time, it is a reasonable assumption because, although possible, the subversion of a repository has not been observed as the preferred approach for propagating malicious packages.

We now define the *monotonicity* and *restricted control* properties.

PROPERTY 1 (MONOTONICITY). *A feature $x \in X$ is said to be monotonic if and only if $x$ is a numerical feature, and any update on its value, $x.value$, can only occur in one direction.*

For example, if *package_age* is a feature (measured in years) and this value can only be increased, we say that *package_age* possesses monotonicity property. On the other hand, package *description_length* as a feature can be increased or decreased by the author of the package and is thus non-monotonic. The monotonicity property is hereinafter referred to as *Property 1*.

PROPERTY 2 (RESTRICTED CONTROL). *A feature $x \in X$ is said to possess the property of restricted control if and only if a stakeholder in $S_i$ associated with package $p_i$ cannot change its value, $x.value$.*

For example, if *number_of_stars* is a feature (measured in count). This feature is calculated based on the interactions that other developers and code users interact with a given package. As such *number_of_stars* cannot directly be modified by a package author. Thus, we say that *number_of_stars* possesses the property of restricted control. A counter-example is *number_of_versions*, which a package author can directly influence by generating several versions. In this case, we say that *number_of_versions* possesses the monotonicity property but lacks the property of restricted control. The restricted control property is hereinafter referred to as *Property 2*.

We define a feature $x \in X$, specially denoted by $\bar{x}$, as a **difficult-to-manipulate** (DTM) feature if any one of the following cases holds:

(1) If $x$ satisfies the monotonicity property i.e. $x \asymp (Property\ 1)$, then $x := \bar{x}$;
(2) If $x$ satisfies the restricted control property i.e. $x \asymp (Property\ 2)$, then $x := \bar{x}$;

Otherwise, $x$ is considered an **easy-to-manipulate** (ETM) feature. It is important to note that $X$ comprises both easy-to-manipulate (ETM) and difficult-to-manipulate (DTM) features denoted by $x$ and $\bar{x}$ respectively i.e. $X \ni x, \bar{x}$.

## 5.1 Easy-to-Manipulate Features

As noted earlier, an easy-to-manipulate feature denoted by $x$ is a feature that does not possess either Property 1 or Property 2 and thus can easily be changed by the author of a package. Although ETM features are inherently good at helping to predict malicious packages (*c.f.* Section 6.2), by being able to manipulate these features, an adversary can *trick* detection models to classify malicious packages as benign. In our metadata feature extraction $\mathcal{F}$, we identify the following types of features as not satisfying either Property 1 or Property 2 and thus considered as ETM.

- **Exist**: This type of feature refers to whether or not certain Information is present in package metadata. This takes on a binary indicator whose value is *TRUE* or *FALSE* depending on whether or not the specified Information is present.
- **Special Character**: A special character is any character that is not a letter, digit, or whitespace. The use of special characters in package names is known to be indicative of typo-squatting [24, 27].
- **Length**: The length of an item is the number of characters it contains, can serve as a useful indicator of malicious packages, especially when they lack detailed descriptions.

Our experiments show that, although these types of features are simple and easy-to-manipulate by the adversary, they are often useful predictors of maliciousness. For example, if the metadata of a package does not contain author information or source code address, that package is likely to be malicious. However, models

**Table 4: List of easy-to-manipulate (ETM) and difficult-to-manipulate (DTM) Features**

| ETM Features | DTM Features |
|---|---|
| name_exist, name_length, dist-tags_exist, dist-tags_length, versions_exist, versions_length, versions_num_count, maintainers_exist, description_exist, description_length, readme_exist, readme_length, scripts_exist, scripts_length, author_exist, author_name, author_email, License_exist, License_length, directories_exist, directories_length, keywords_exist, keywords_length, keywords_num_count, homepage_exist, homepage_length, github_exist, github_length, bugslink_exist, bugslink_length, issueslink_exist, issueslink_length. dependencies_exist, dependencies_length, devDependencies_exist, devDependencies_length | package_age, package_modified_duration, package_published_duration, author_CPN, author_service_time, author_CCS, maintainer_CPN, maintainer_service_time, maintainer_CCS, contributor_CPN, contributor_service_time, contributor_CCS, publisher_CPN, publisher_service_time, publisher_CCS, pull_request, issues, fork_number, star, subscriber_count |

[*] CCS means community contribution score and CPN means contribute package number.

built solely on these features are vulnerable to adversarial attacks. Incorporating DTM features can mitigate the risk.

## 5.2 Difficult-to-Manipulate Features

These are features which satisfy Property 1 or 2. They often depend on time or package interaction, which are difficult to manipulate. The types of features in this category are as follows:

- **Temporal**: Features that involve temporal information often satisfy Property 1 and as such are DTM. In this work, our feature extractor $\mathcal{F}$ generates *package_age*, *package_modified_duration* and *package_published_duration* which represent the age of the package, the time interval between package creation and last modification date, and the time interval between when the package was created and when it was published respectively. Other features include stakeholder $s_{j_i}$ service time ($s_{j_i}\_service\_time$) which reflects the number of days which a stakeholder has been associated with the package $p_i$.

- **Package Interaction**: This relates to the number of interactions that a package $p_i$ or its stakeholder $s_{j_i}$ has. It includes (1) number of other packages which $s_{j_i}$ has contributed to denoted $s_{j_i}\_CPN$; (2) number of package pull requests *pull_request*; (3) number of reported package issues; (4) number of times package is forked; and (5) number of stars a package has received. (1) satisfies Property 1 while (2), (3), (4) and (5) satisfy Property 2.

Table 4 provides the list of the ETM and DTM features used in this work. It is worth noting that the DTM features in the table also include a combination of base DTM features *e.g.* stakeholders' community contribution score ($s_{j_i}\_CCS$) is a combination of stakeholder contribute package number $s_{j_i}\_CPN$ and stakeholder service time $s_{j_i}\_service\_time$. Appendix A.1 provides details for $s_{j_i}\_CCS$ DTM features derived from base DTM features.

## 5.3 Proposed *MeMPtec* Model

Figure 1 shows the pipeline for our proposed Metadata based Malicious Package Detection (*MeMPtec*) model. The figure shows the phases of model building *i.e.* training phase and prediction phase. In the training phase, PMI is fed into the feature extraction stage and assigned a label as either benign or malicious. The metadata is extracted using our feature extractor $\mathcal{F}$ into both easy-to-manipulate

(ETM) (*c.f. Section 5.1*) and difficult-to-manipulate (DTM) (*c.f. Section 5.2*) features. We then adopt existing machine and deep learning models to train a model. In the prediction phase, we follow a similar process of feature extraction, feeding these extracted features into the built model to make predictions regarding the maliciousness of packages.

Algorithm 1 gives the details of the steps in *MeMPtec*. It takes PMI $\{\mathcal{I}_1, \cdots, \mathcal{I}_n\}$, ML_Algo, $\{\mathcal{I}_{new}\}$ as input and provides malicious package detector $\mathcal{M}$ as output. The algorithm has two parts: Model Training Phase and Prediction Phase. In the Training Phase, we extract labels Y, ETM and DTM features (*c.f. Section 5.1 and 5.2*) in lines 3-5, respectively. Then, we combine two sets of features and create X in line 6. The X and Y are partitioned into train data (70%), validation data (10%) and test data (20%) in line 7. After that, the model is built based on the existing machine learning algorithm and train and validation data in line 8. The build-in model $\mathcal{M}$ performance has been measured using test data in lines 9-10. Therefore, the model training phase returns malicious package detector model $\mathcal{M}$ and performance in line 11.

In the prediction phase, we similarly extract the relevant features and apply the built model $\mathcal{M}$ to each set of features $X_{new}$ associated with a package's PMI $\mathcal{I}_{new}$ (lines 13–16). The function returns predicted label in line 17. Finally, in lines 18 and 19, these two phases are called to as model training and prediction.

## 6 EXPERIMENTS

### 6.1 Experimental Setup

It is worth recalling that the crux of this work is in its feature engineering approach; thus we compare our approach with existing features proposed by closely related work such as [1, 15, 26, 31, 36]. All experiments were implemented in Python and conducted in Windows 10 environment, on an Intel Core i7 processor (1.70 GHz, 16GB RAM).

*6.1.1 Datasets and Baseline Methods.* In this work, we use NPM repository [2] as an exemplar to generate package metadata information. We make the assumption that packages that are currently not flagged as malicious in NPM repository are considered benign. In NPM repository, packages flagged as malicious are often removed. Thus, to obtain malicious NPM packages, we use publicly available data on GitHub [3] [16]. We then generate a balanced

---
[2]https://registry.npmjs.org/
[3]https://dasfreak.github.io/Backstabbers-Knife-Collection

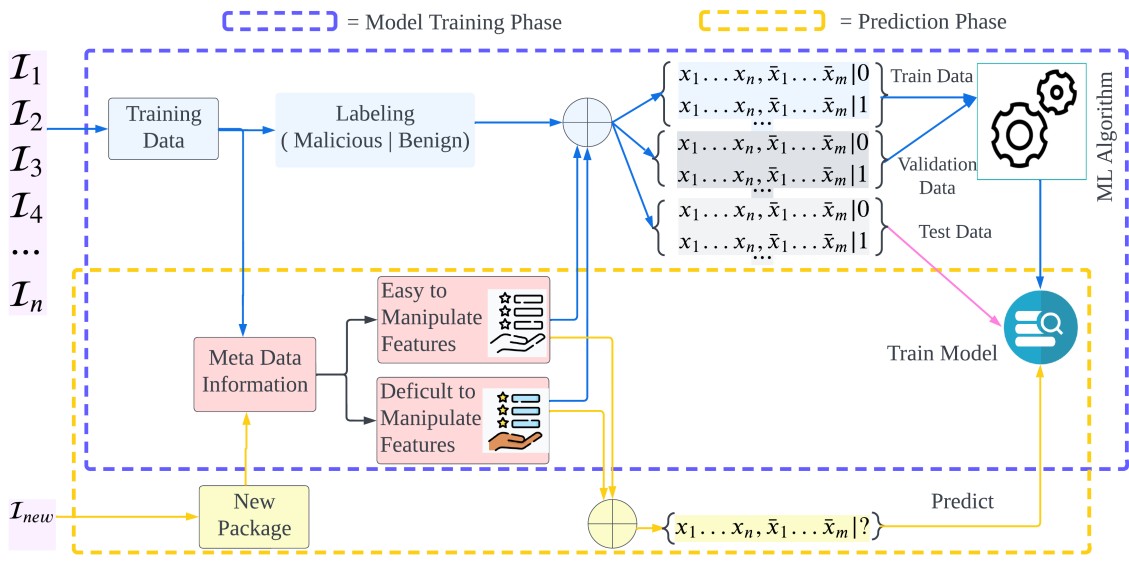

**Figure 1: Proposed Metadata-based Malicious Package Detection (*MeMPtec*) model architecture.**

---

**Algorithm 1:** MeMPtec($\{\mathcal{I}_1, \cdots, \mathcal{I}_n\}$, ML_Algo, $\{\mathcal{I}_{new}\}$)

**Data:** $\{\mathcal{I}_1, \cdots, \mathcal{I}_n\}$ : Label packages metadata information
1 ; *ML_Algo* : ML / DL algorithm; $\{\mathcal{I}_{new}\}$ : New package;
   **Result:** Malicious Package Detector $\mathcal{M}$
2 **Function** Model_Training_Phase($\{\mathcal{I}_1, \cdots, \mathcal{I}_n\}$, *ML_Algo*):
3     Y ← Extract label (Malicious or Benign) from $\{\mathcal{I}_{new} \cdots, \mathcal{I}_n\}$.
4     ETM_Features ($\{x_1 \cdots, x_n\}$) ← Extract easy to manipulate features
       (*c.f.* Section 5.1) from $\{\mathcal{I}_1, \cdots, \mathcal{I}_n\}$.
5     DTM_Features ($\{\bar{x}_1 \cdots \bar{x}_m\}$) ← Extract difficult to manipulate features
       (*c.f.* Section 5.2) from $\{\mathcal{I}_1, \cdots, \mathcal{I}_n\}$.
6     X ← ETM_Features ⊕ DTM_Features
7     $X_{train}, X_{valid}, X_{test}, Y_{train}, Y_{valid}, Y_{test}$ ← split(X,Y, 0.7, 0.1, 0.2)
8     $\mathcal{M}$ ← Build_Model( ML_Algo, $X_{train}, X_{valid}, Y_{valid}, Y_{test}$ )
9     Predict_Test_Result ← $\mathcal{M}$.predict( $X_{test}$ )
10     Performance ← Performance_Measurement(Predict_Test_Result, $Y_{test}$)
11     **Return:** $\mathcal{M}$, Performance
12 **Function** Prediction_Phase($\{\mathcal{M}, \{\mathcal{I}_{new}\}$):
13     $ETM\_Features_{new}$ ($\{x_1 \cdots, x_n\}$) ← Extract easy to manipulate
       features (*c.f.* Section 5.1) from $\{\mathcal{I}_{new}\}$.
14     $DTM\_Features_{new}$ ($\{\bar{x}_1 \cdots \bar{x}_m\}$) ← Extract difficult to manipulate
       features (*c.f.* Section 5.2) from $\{\mathcal{I}_{new}\}$.
15     $X_{new}$ ← $ETM\_Features_{new} \oplus DTM\_Features_{new}$
16     Predict_label ← $\mathcal{M}$.predict($X_{new}$)
17     **Return:** Predict_label
18 $\mathcal{M}$, Performance ← Model_Training_Phase($\{\mathcal{I}_1, \cdots, \mathcal{I}_n\}$, *ML_Algo*)
19 Predict_label ← Prediction_Phase($\mathcal{M}, \{\mathcal{I}_{new}\}$)

---

dataset with a proportion of 50% malicious packages, and an imbalanced datasets with only 10% malicious packages. Variants of these datasets are generated for experimental purposes (Table 5). In the table, *Existing_tec* refers to feature model generated using features proposed in the literature [1, 15, 26, 31, 36]; *MeMPtec_E* and *MeMPtec_D* refer to feature model with ETM and DTM features respectively; while *MeMPtec* refers to the combination of ETM and DTM features based feature model.

*6.1.2 Machine Learning/Deep Learning Techniques.* In building the detection models, we adopted five different but commonly used

**Table 5: Description of datasets parameters.**

| Feature Model | # Features | Balance Data | | Imbalance Data | |
|---|---|---|---|---|---|
| | | # Malicious | # Benign | # Malicious | # Benign |
| *Existing_tec* | 11 | 3232 | 3232 | 3232 | 32320 |
| *MeMPtec_E* | 36 | 3232 | 3232 | 3232 | 32320 |
| *MeMPtec_D* | 21 | 3232 | 3232 | 3232 | 32320 |
| *MeMPtec* | 57 | 3232 | 3232 | 3232 | 32320 |

model building techniques namely, *Support Vector Machine* [18]; *Gradient Boosting Machine (GBM)* [4]; *Generalized Linear Model (GLM)* [11]; *Distributed Random Forest (DRF)* [7]; and *Deep Learning - ANN (DL)* [2, 33].

In all experiments, we adopt a 70:10:20 split for training, validation and testing, respectively, and conduct five-fold cross-validation.

*6.1.3 Evaluation Metrics.* In this work, we adopt the well-known metrics of *precision, recall, F1-score, accuracy* and *root mean squared error* (RMSE) also used in [8, 15]. We also evaluate model performances based on the number of *false positives* (FP) and *false negatives* (FN) like in [19, 20].

## 6.2 Performance Evaluation of *MeMPtec* (RQ1)

Table 6 shows the performance analysis of our proposed approach. From the table, we notice that *MeMPtec* (*resp.* balance and imbalance data) consistently achieves the best results across all metrics and ML/DL algorithms. It is important to note that *RMSE* indicates the confidence of a model in its prediction as it measures the error between the probability of the prediction and the true label. Notice that *MeMPtec* (*resp.* for both data) consistently has significantly lower errors, indicating that combining ETM and DTM leads to more robust model.

Although one may question the significance of the improvement, it is important to note that in the domain of software security, marginal improvements are desirable since even 1 missed malicious

**Table 6: Performance evaluation results in terms of the mean and standard errors: ↑ (resp. ↓) indicate higher (resp. lower) results are better; bold values represent the best result and underlined values represent the second best result.**

| | ML/DL Algo | Feature Model | Precision ↑ | Recall ↑ | F1-score ↑ | Accuracy ↑ | RMSE ↓ |
|---|---|---|---|---|---|---|---|
| **Balance Data** | SVM | *Existing_tec* | 0.9651 ± 0.003 | 0.9817 ± 0.002 | 0.9733 ± 0.001 | 0.9731 ± 0.001 | 0.1640 ± 0.002 |
| | | *MeMPtec_E* | **0.9994 ± 0.000** | 0.9725 ± 0.003 | 0.9857 ± 0.002 | 0.9859 ± 0.002 | 0.1175 ± 0.008 |
| | | *MeMPtec_D* | 0.9856 ± 0.002 | **0.9972 ± 0.001** | 0.9914 ± 0.001 | 0.9913 ± 0.001 | 0.0927 ± 0.004 |
| | | *MeMPtec* | 0.9960 ± 0.002 | 0.9963 ± 0.001 | **0.9962 ± 0.002** | **0.9961 ± 0.002** | **0.0576 ± 0.012** |
| | GLM | *Existing_tec* | 0.9798 ± 0.001 | 0.9734 ± 0.002 | 0.9766 ± 0.001 | 0.9766 ± 0.001 | 0.1595 ± 0.002 |
| | | *MeMPtec_E* | 0.9875 ± 0.003 | 0.9817 ± 0.003 | 0.9846 ± 0.002 | 0.9847 ± 0.002 | 0.1032 ± 0.006 |
| | | *MeMPtec_D* | 0.9951 ± 0.001 | 0.9963 ± 0.001 | 0.9957 ± 0.000 | 0.9957 ± 0.000 | 0.0689 ± 0.002 |
| | | *MeMPtec* | **0.9997 ± 0.000** | **0.9969 ± 0.000** | **0.9983 ± 0.000** | **0.9983 ± 0.000** | **0.0395 ± 0.005** |
| | GBM | *Existing_tec* | 0.9813 ± 0.001 | 0.9753 ± 0.002 | 0.9783 ± 0.001 | 0.9783 ± 0.001 | 0.1407 ± 0.003 |
| | | *MeMPtec_E* | 0.9966 ± 0.001 | 0.9947 ± 0.001 | 0.9956 ± 0.001 | 0.9957 ± 0.001 | 0.0581 ± 0.006 |
| | | *MeMPtec_D* | 0.9963 ± 0.002 | 0.9976 ± 0.001 | 0.9969 ± 0.001 | 0.9969 ± 0.001 | 0.0512 ± 0.004 |
| | | *MeMPtec* | **0.9997 ± 0.000** | **0.9988 ± 0.001** | **0.9992 ± 0.000** | **0.9992 ± 0.000** | **0.0321 ± 0.004** |
| | DRF | *Existing_tec* | 0.9798 ± 0.001 | 0.9762 ± 0.003 | 0.9780 ± 0.001 | 0.9780 ± 0.001 | 0.1416 ± 0.003 |
| | | *MeMPtec_E* | 0.9982 ± 0.001 | 0.9941 ± 0.002 | 0.9961 ± 0.001 | 0.9961 ± 0.001 | 0.0548 ± 0.006 |
| | | *MeMPtec_D* | 0.9963 ± 0.002 | 0.9972 ± 0.001 | 0.9968 ± 0.000 | 0.9968 ± 0.000 | 0.0471 ± 0.002 |
| | | *MeMPtec* | **0.9991 ± 0.001** | **0.9997 ± 0.000** | **0.9994 ± 0.000** | **0.9994 ± 0.000** | **0.0225 ± 0.002** |
| | DL | *Existing_tec* | 0.9810 ± 0.001 | 0.9756 ± 0.002 | 0.9783 ± 0.001 | 0.9783 ± 0.001 | 0.1447 ± 0.003 |
| | | *MeMPtec_E* | 0.9891 ± 0.003 | 0.9922 ± 0.002 | 0.9907 ± 0.002 | 0.9907 ± 0.002 | 0.0874 ± 0.011 |
| | | *MeMPtec_D* | 0.9954 ± 0.002 | 0.9969 ± 0.000 | 0.9961 ± 0.001 | 0.9961 ± 0.001 | 0.0597 ± 0.007 |
| | | *MeMPtec* | **0.9981 ± 0.001** | **0.9988 ± 0.001** | **0.9984 ± 0.001** | **0.9985 ± 0.001** | **0.0288 ± 0.009** |
| **Imbalance Data** | SVM | *Existing_tec* | 0.9127 ± 0.004 | 0.9511 ± 0.006 | 0.9314 ± 0.004 | 0.9873± 0.001 | 0.1126 ± 0.003 |
| | | *MeMPtec_E* | 0.9940 ± 0.001 | 0.9688 ± 0.003 | 0.9812 ± 0.001 | 0.9966 ± 0.000 | 0.0579 ± 0.002 |
| | | *MeMPtec_D* | 0.9799 ± 0.004 | 0.9417 ± 0.014 | 0.9601 ± 0.006 | 0.9929 ± 0.001 | 0.0833 ± 0.006 |
| | | *MeMPtec* | **0.9981 ± 0.001** | 0.9765 ± 0.003 | **0.9872 ± 0.001** | **0.9977 ± 0.000** | **0.0477 ± 0.003** |
| | GLM | *D_imb_L* | 0.9134 ± 0.010 | 0.9508 ± 0.014 | 0.9317 ± 0.008 | 0.9873 ± 0.001 | 0.1094 ± 0.005 |
| | | *MeMPtec_E* | **0.9981 ± 0.002** | 0.9688 ± 0.007 | 0.9832 ± 0.003 | 0.9970 ± 0.001 | 0.0559 ± 0.005 |
| | | *MeMPtec_D* | 0.9776 ± 0.004 | 0.9663 ± 0.006 | 0.9718 ± 0.004 | 0.9949 ± 0.001 | 0.0712 ± 0.003 |
| | | *MeMPtec* | 0.9970 ± 0.001 | **0.9848 ± 0.002** | **0.9909 ± 0.001** | **0.9983 ± 0.000** | **0.0361 ± 0.001** |
| | GBM | *D_imb_L* | 0.9208 ± 0.003 | 0.9502 ± 0.007 | 0.9352 ± 0.003 | 0.988 ± 0.001 | 0.1000 ± 0.002 |
| | | *MeMPtec_E* | 0.9927 ± 0.002 | 0.9870 ± 0.003 | 0.9898 ± 0.002 | 0.9982 ± 0.000 | 0.0392 ± 0.004 |
| | | *MeMPtec_D* | 0.9905 ± 0.002 | 0.9947 ± 0.001 | 0.9926 ± 0.001 | 0.9986 ± 0.000 | 0.0320 ± 0.003 |
| | | *MeMPtec* | **0.9984 ± 0.001** | **0.9954 ± 0.001** | **0.9969 ± 0.001** | **0.9994 ± 0.000** | **0.0189 ± 0.001** |
| | DRF | *D_imb_L* | 0.9228 ± 0.004 | 0.9511 ± 0.007 | 0.9367 ± 0.003 | 0.9883 ± 0.001 | 0.0991 ± 0.003 |
| | | *MeMPtec_E* | 0.9978 ± 0.001 | 0.9880 ± 0.002 | 0.9929 ± 0.001 | 0.9987 ± 0.000 | 0.0321 ± 0.003 |
| | | *MeMPtec_D* | 0.9932 ± 0.001 | 0.9931 ± 0.003 | 0.9931 ± 0.002 | 0.9988 ± 0.000 | 0.0322 ± 0.003 |
| | | *MeMPtec* | **0.9979 ± 0.001** | **0.9984 ± 0.001** | **0.9981 ± 0.000** | **0.9997 ± 0.000** | **0.0185 ± 0.001** |
| | DL | *D_imb_L* | 0.9221 ± 0.004 | 0.9502 ± 0.007 | 0.9359 ± 0.003 | 0.9882 ± 0.001 | 0.1005 ± 0.002 |
| | | *MeMPtec_E* | 0.9907 ± 0.003 | 0.9793 ± 0.004 | 0.9849 ± 0.002 | 0.9973 ± 0.000 | 0.0471 ± 0.004 |
| | | *MeMPtec_D* | 0.9877 ± 0.004 | 0.9907 ± 0.003 | 0.9891 ± 0.002 | 0.9980 ± 0.000 | 0.0429 ± 0.005 |
| | | *MeMPtec* | **0.9982 ± 0.001** | **0.9966 ± 0.001** | **0.9974 ± 0.001** | **0.9995 ± 0.000** | **0.0209 ± 0.003** |

package (false negative) can have catastrophic consequences. For this reason, we further analyse the false positives (FP) and false negatives (FN). In a balanced dataset, *MeMPtec* significantly outperforms *Existing_tec* in reducing FP in Figure 3 (a). On the GLM algorithm, *MeMPtec* achieves a 98.33% reduction (12.0 → 0.2), and on the SVM algorithm, it achieves an 88.69% reduction (23.0 → 2.6). On average, *MeMPtec* reduces FPs by 93.44% (14.64 → 0.96). *MeMPtec* also performs well in reducing FP in Figure 3 (b). It reduces the maximum number of FNs by 98.70% on the DRF algorithm (15.4 → 0.2) and achieves a minimum reduction of 79.66% (11.8 → 2.4) on

the SVM algorithm. On average, *MeMPtec* reduces FNs by 91.86% (15.24 → 1.24).

The results in Figure 3 (c) exhibit consistent trends in the imbalanced dataset. *MeMPtec* reduces FP maximum 97.96% (58.8 to 1.2) on SVM, minimum 97.29% (51.4 → 1.4) on DRF algorithm. It reduces FP on average 97.5% (54.6 → 1.36) than the *Existing_tec*. It also reduces, on average, 80.42 % of the FN numbers from 31.88 → 6.24 in Figure 3 (d). In all Figures 3, we observe that by using *MeMPtec_E* and *MeMPtec_D*, the FP and FN can be reduced by an order of magnitude than the *Existing_tec*. These experiments illustrate the efficacy of *MeMPtec* in addressing **Challenges 1 & 3**.

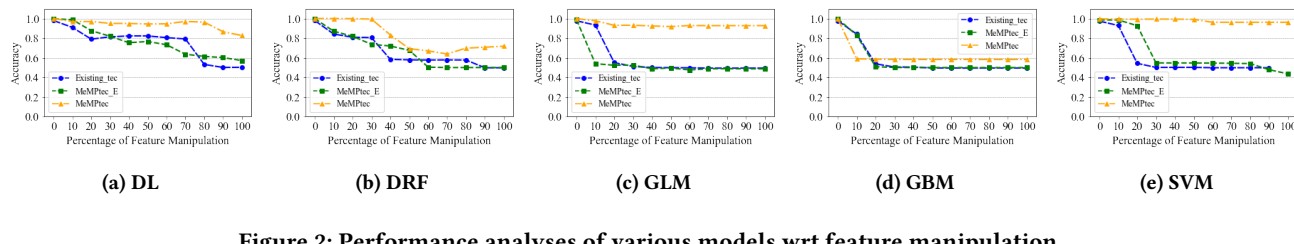

**(a) DL**  **(b) DRF**  **(c) GLM**  **(d) GBM**  **(e) SVM**

Figure 2: Performance analyses of various models wrt feature manipulation.

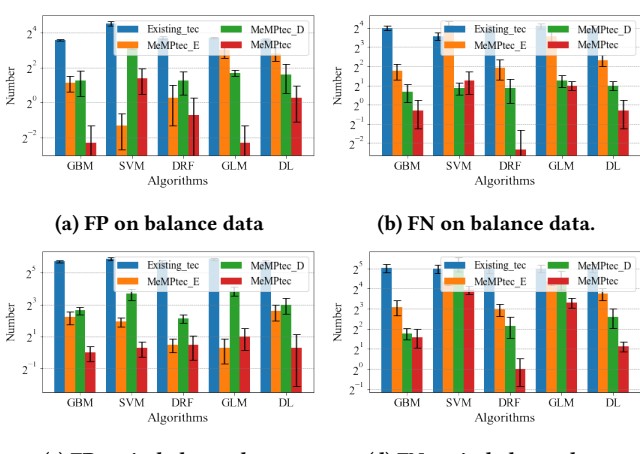

**(a) FP on balance data**  **(b) FN on balance data.**

**(c) FP on imbalance data**  **(d) FN on imbalance data**

Figure 3: False Positive and False Negative numbers comparison on balanced and imbalanced datasets.

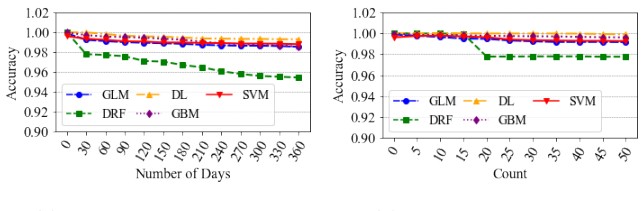

**(a) Temporal Information**  **(b) Package Interaction**

Figure 4: Performance analyses of *MeMPtec* wrt monotonic property (temporal information and package interaction).

## 6.3 Robustness of *MeMPtec* (RQ2)

In this section, we evaluate the robustness of *MeMPtec* against adversarial attack. We assess the impact of data manipulation on the performance of the models by (1) ranking the features for each dataset according to their importance for each model; and (2) replacing the true values of the features in the malicious dataset with random values selected from a distribution of values for the same feature in the benign dataset iteratively beginning from the most important feature (Appendix A.2 has the details of the algorithm). By doing this, we are simulating various degrees of the worst-case scenario adversarial attack where an adversary deliberating tries to game the model.

Figure 2 is the result of this experiment. In this experiment, in decreasing order of importance, the values of features for the malicious dataset are replaced. The figure shows the decline in accuracy performance for the balanced dataset across the models. We note that in all the models, as the percentage of features is manipulated, the model performance decreases drastically for the *Existing_tec* and *MeMPtec_E* based features. However, this is less so for the *MeMPtec* features. In fact, even after manipulating 100% of the features *MeMPtec* based approach performs significantly better (*e.g* GLM model: 99.87% → 92.73%). We conduct further extensive experiments, achieving similar results, by considering only the top ten features (Appendix A.3) as well as indirect manipulation of the features via the package metadata information (Appendix A.3)– not included due to space constraints.

In Figure 4, we also investigate the impact of the monotonicity property on the ability of an adversary to manipulate the DTM features. Figure 4 (a) shows the modification of all temporal DTM features by increasing their time-based values iteratively (in number of days). The aim of the experiment is to show the robustness of *MeMPtec* even when the adversary attempts to game the model via DTM features. We note that for DL, even after 360 days, *MeMPtec* features only decline marginally in performance (0.9998 → 0.9928). Similarly, Figure 4 (b) shows the modification of all package interaction-based DTM features. In this experiment, the count of each figure increased iteratively. Similarly, we notice that for DL, *MeMPtec* features only decline marginally in performance after 50 count updates (0.9998 → 0.9989). As can be seen, the behaviour is consistent across all the different models.

These experiments validate the *MeMPtec*'s ability to mitigate against adversarial attacks (**Challenge 2**).

## 7 CONCLUSION

In this paper, we proposed metadata based malicious detection algorithm named *MeMPtec*, which relies on a novel feature engineering strategy resulting in easy-to-manipulate (ETM) and difficult-to-manipulate (DTM) features from metadata. We conduct extensive experiments to demonstrate *MeMPtec*'s efficacy for detecting malicious packages in comparison with existing approaches proposed in the state-of-the-art. In particular, *MeMPtec* leads to an average reduction of false positives by an impressive 93.44% and 97.5% across two experimental datasets, respectively. Additionally, false negative numbers decrease on average 91.86% and 80.42% across the same datasets, respectively. Furthermore, we analyse *MeMPtec*'s resistance against adversarial attacks and show that, even under worst-case scenarios, our approach is still highly resistant.

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

# A APPENDIX

## A.1 Stakeholders Community Contribution Score

Stakeholders play a significant role in ensuring malicious package detection. It has been seen that popular or well-known stakeholders are not involved as intruders. Thus, using the following equation, we can define the stakeholders' community contribution score ($s_{j_i}\_CCS$) using the stakeholder contribute package number and service time for each package $p_i$.

$$s_{j_i}\_CCS = Log_x(s_{j_i}\_service\_time) * Log_x(s_{j_i}\_CPN) \qquad (1)$$

We define the stakeholder's community contribution score based on logarithm base x (x= 2 default). The main reason for this logarithm base score is that we want to avoid a certain label of manipulation. Although it is difficult to manipulate author contributions, it might be possible that attackers can upload multiple packages and increase their stakeholder contribution package number. Thus, we defined the $s_{j_i}$ that stakeholders can not change easily without considering temporal and package interaction properties.

## A.2 MeMPtec Adversarial Manipulation Algorithm

To prevent adversaries, we analysed data manipulation-based performance analysis in the algorithm 2. The algorithm takes build model $\mathcal{M}$ (from algorithm 1) and adversaries data as input and returns adversaries-based results. Initially, we set a data frame that is empty. Then, we calculate features significant for each model and find the significant feature ranked based on Shapley additive explanation (SHAP) [13] values (decreasing order) in line 2.

---

**Algorithm 2:** MeMPtec Adversaries ( $\mathcal{M}$, Data )

**Data:** $\mathcal{M}$: Build *MeMPtec* Model; *Data* : Machine transferable data;
**Result:** DataFrame: Models performance based on data manipulation;
1  DataFrame ← ∅
2  Significant_Feature ← Rank_Features($\mathcal{M}$, SHAP)
3  manipulate_data ← Data.copy()
4  Original_Label ← Extract label from manipulate_data
5  Predict_Result ← $\mathcal{M}$.predict(manipulate_data)
6  Performance ← Performance_Measurement(Predict_Result, label_Test)
7  DataFrame ← DataFrame ∪ [$\mathcal{M}$.name, "ALL', Performance]
8  Number_of_MF = [TOP-N | len(Significant_Feature) if option = TOP-N | Percentage]
9  **for** $i \in range(Number\_of\_MF)$ **do**
10      feature = Significant_Feature[i]
11      manipulate_data = Manipulate_Data(manipulate_data, feature)
12      Predict_label ← $\mathcal{M}$.predict(manipulate_data)
13      Performance ← Performance_Measurement( Predict_label, Original_label)
14      DataFrame ← DataFrame ∪ [$\mathcal{M}$.name, feature, Performance]
15  **Return:** DataFrame          /* Return manipulated feature based results. */

---

Manipulate data has been initialised by our machine transferable data in line 3. Then, we extract the original label that should be used to check our predicted results accuracy measurement in line 4 and find the without manipulated data-based results in line 5. We measure the results and save them in the data frame in lines 6 and 7, respectively. This model is applicable for TOP-N feature manipulation analysis as well as percentage of features manipulation analysis. Thus, we select the number of manipulated features in line

8, where TOP-N selects only TOP-N features and the percentage option selects all feature numbers. In the feature item, we only manipulate corresponding malicious package feature values based on benign value distributions in lines 10-11. After that, predict the target variable using manipulated data and the selected model in line 12. Furthermore, various evaluation metric values have been calculated using prediction and original output and saved to the data frame in lines 13 and 14, respectively. This process continues for each feature in the model and each model in our considered five ML/DL methods. Finally, the algorithm returns the manipulated results for TOP-N or Percentage in line 15.

## A.3 TOP-N Features Manipulation Analysis

Generally, the attacker's motive is to manipulate less number of features that have a significant influence on the model performance degradation. To consider this intention, we analysis the performance of our features-based algorithm considering TOP-N significant information and features. To detect the significant features, we used SHAP [13] values ranking algorithms.

Figure 5 shows the TOP-10 features manipulation result performance. It is clear that our *MeMPtec* based results are more robust than the *MeMPtec_E* and *Existing_tec* for all algorithms. The main reason is the significant features that each algorithm selects based on its dataset. In our proposed feature selection method *MeMPtec*, top notable features are difficult to manipulate that attackers can not change easily. As a result, the model performance reduces a little bit. For example, after 10 features manipulation, *MeMPtec* performance reduces 99.94% → 89.55% in DL, 99.98% → 99.98% in DRF 98.87% → 95.25% in GLM, 99.95% → 58.16% in GBM and 99.59% → 99.13 in SVM model. In contrast, *Existing_tec* based features performance reduced rapidly and reached around 50.0% for all ML/DL methods.

## A.4 Information Manipulation Analysis

In this research work, we have utilised information and feature. Thus, we can easily modify algorithm 2 for information manipulation. To make the algorithm for information manipulatable, we make information ranked based on their features SHAP values. After that, we change that information one by one by changing their corresponding features manipulation and find the results.

We observe similar results patterns in figure 2 for the percentage of information manipulation in Figure 6. In the GLM algorithm, *MeMPtec* information reduces model performances by 7.19% (99.87% → 92.68%) after 100% manipulation, while *Existing_tec* information reduces model performances by around 46.70% (97.64% → 50.94%) after only 30% information manipulation. In the DL algorithm, *MeMPtec* based performance reduces 17.32% (99.98% → 82.66%), whereas *Existing_tec*-based performance reduces 47.82% (97.94% → 50.12%).

Figure 7 shows the TOP-N (1-10) significant information changed based on results. This result is slightly different from the TOP-N features results because, in this case, we added corresponding features SHAP values to indicate information SHAP values. That means the selected information set differs from the chosen TOP-N features set. Our *MeMPtec* based results outperform the *MeMPtec_E* and *Existing_tec* for all algorithms regarding model robustness. For

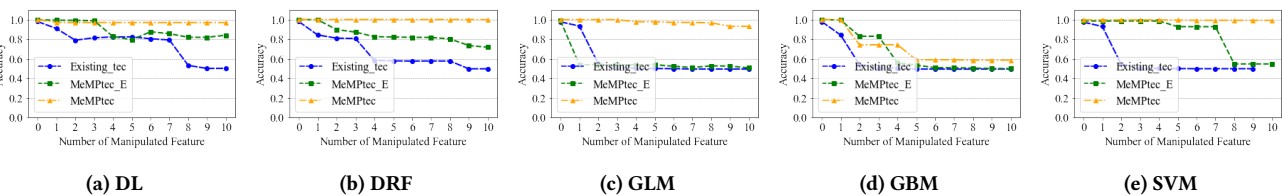

Figure 5: Performance analyses of various models wrt TOP-N significant feature manipulation.

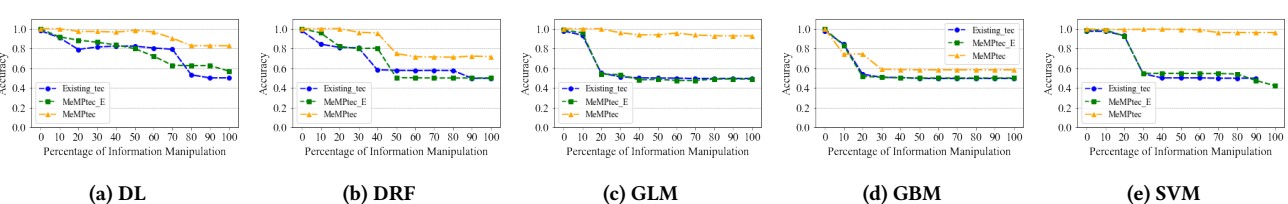

Figure 6: Performance analyses of various models wrt information manipulation.

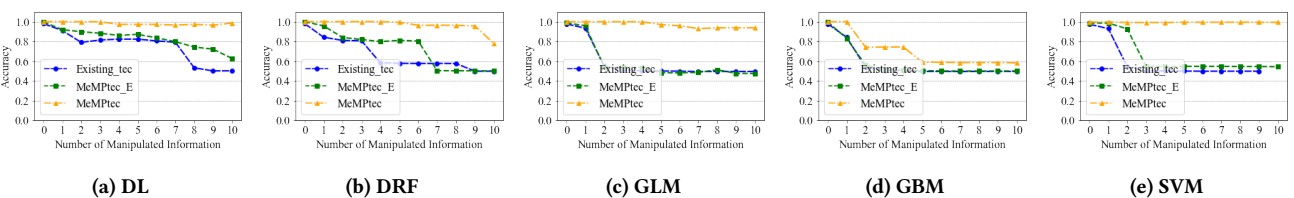

Figure 7: Performance analyses of various models wrt TOP-N significant information manipulation.

example, after 10 information manipulation *MeMPtec* method performances reduced 99.94% → 81.03% in DL, 99.98% → 99.98% in DRF and 98.87% → 93.25% in GLM, whereas *Existing_tec* based features performances reduced rapidly and reached around 50.0% for all algorithms. It shows that *MeMPtec* performances reduce

significantly on the GBM algorithm and it is still better than the *Existing_tec* model. Finally, we can say our *MeMPtec* feature selection model outperforms existing works for well known machine learning algorithms.

