# OpenReview forum: "Malicious Package Detection using Metadata Information"
_ACM.org/TheWebConf/2024/Conference — TheWebConf24_

### Official Review · Reviewer_JnBT · 2023-11-13

**Novelty:** 5
**Technical Quality:** 5

**Review:**

This work focuses on the big, unsolved problem of supply-chain security. Indeed, especially in the context of open source, third party libraries and applications are commonly used in software development. This poses a significant threat for developers that might include in their code base unwanted software such as operational libraries exhibiting a malicious behavior. To cope with this issue, the authors introduce a novel detection model that leverages metadata information to infer the reputation of a third party package. The model is tested against adversarial attacks and compared with previous work.

The paper is well written and easy to follow. The authors did a good job in motivating their work and discussing their contributions in this research field. While other researchers had previously suggested using metadata information to infer the reputation of a software package, this work extends on previous research. The authors introduced a novel set of features (i.e., temporal and interaction in Table 1) and did a meticulous work in feature engineering (the main contribution) that resulted in a model which proved to outperform previous work (as per Table 6). However, there are some limitations and considerations that I would like to mention. See the questions section.

**Questions:**

- Figure 2 shows the resilience of the model to adversarial attacks. For example, a) depicts how the accuracy of the model decreases with the increase on the percentage of features manipulation. Could the authors explain why their best model (in yellow, MeMPtec) still performs 80% detection when all the features (100% rate) are actually manipulated? [Rebuttal update: this point will be better elaborated in the camera ready version of the paper]
- To have a better feeling of the prevalence of packages with metadata shady information, it would be interesting if the authors could give some numbers (based on their experience) of how many malicious packages are actually found in the wild out of the total, e.g. on a daily basis.
- How many of these packages are actually malicious by nature, or are actually benign packages being hijacked? In this case, I assume the model won't be able to detect them because the metadata would were not changed. Could the authors better discuss this point?
- The paper is lacking a discussion on the example of some False Positive. Why these occur and how the model could be improved for further reduction?
- Do the author plan to introduce a novel class, e.g. suspicious, to handle suspicious packages, e.g. packages with generic metadata information that would might fool the detection?
- s/true/original on line 6 of Section 6.3

**Ethics Review Description:**

Nothing

**Reviewer Confidence:**

3: The reviewer is confident but not certain that the evaluation is correct

**Scope:**

3: The work is somewhat relevant to the Web and to the track, and is of narrow interest to a sub-community

---

### Official Review · Reviewer_8yKD · 2023-11-19

**Novelty:** 3
**Technical Quality:** 3

**Review:**

This work aims to identify malicious NPM packages by using two groups of features, those are easy to manipulate and those are difficult to manipulate, and made a comparison with the model with features used in existing work (for detection tasks among software packages). The used dataset consists of benign NPM packages and malicious ones obtained leveraging a Github project. The experiment results show that (1) the trained model improved the performance by 1 - 3 percents, compared with the basedline model using features from existing work, and (2) the difficult-to-manipulate features plays a significant role.

Pros:
1. A large-enough dataset.

Cons:
1. This work puts a lot of efforts on the feature selection, and thus its research contribution is unclear.
2. Regarding the feature selection,
    1. since the SVM model with the difficult-to-manipulate features only works pretty good (in Table 6, Figures 2, 5, 6, and 7), is it necessary to come up the easy-to-manipulate features?
    2. is it necessary to separate name_exist from name_length, as an example, in Table 4? We can set name_length to -1 to represent name_not_exist.
3. It is unfair to compare the proposed model with the model using features from existing work [1, 15, 26, 31, 36], since many of them are not for the same purpose as this work. In addition, it is better to list the features used in the baseline model.
4. In Algorithm 2 (Appendix A.2), it seems that when M.predict(a'), a is still in the model M, where a is the original data and a' is the manipulate data. This approach isn't realistic. It is better to remove a from M when M.predict(a').
5. Minor mistakes in paper writing.
    1. In line 628, it should not be "10% malicious packages" but 1/11 according to Table 5.
    2. In line 749, the "FP in Figure 3 (b)" should be "FN in Figure 3 (b)".

**Questions:**

Concerns raised up in cons.

**Reviewer Confidence:**

4: The reviewer is certain that the evaluation is correct and very familiar with the relevant literature

**Scope:**

3: The work is somewhat relevant to the Web and to the track, and is of narrow interest to a sub-community

---

### Official Review · Reviewer_P2EH · 2023-11-23

**Novelty:** 4
**Technical Quality:** 6

**Review:**

I thank the authors for this submission, this paper is well written, well motivated and easy to follow. The authors present a new ML methodology for detecting malicious NPM packages, which uses a set of features that are highly resilient to modifications by malicious actors. They show that their models outperform relevant previous work. My main questions and recommendations for this paper are the following:

- Intro: It would be good to add some numbers about the popularity of PyPi, similar to those shown for NPM. If PyPi is not as relevant, then it could also just be mentioned as the Python alternative to NPM.

- The categories presented in Table 1 are different to those presented in S4. They should either be introduced further in S2 or at least be consistent with S4 so that readers can find a definition of these categories.

- Authors assume packets on NPM are not malicious. This is probably true for the vast majority of packages, but it would be good if they could provide any indication of how often malicious packages are found and how long they often survive before being removed.

- Is metadata on NPM packages self-reported by developers or is there some sort of verification process? This should be further developed on this paper, as relying on self-reported metadata haves its own set of inherent challenges. It becomes clearer in S5 that an adversarial attacker can modify those, but it is unclear until then.

- On a similar note, I thin there's a lack of discussion on how likely it is that malicious actors do change these values. The authors show clear examples as to why an actor would do this, but it is unclear if there is any evidence of this sort of behavior happening on the wild.

**Questions:**

- Authors shows that even manipulating 100% of features shows small decreases in accuracy and recall. This begs the question of how needed are these features and whether it would make sense to only use those that are more resilient.

**Reviewer Confidence:**

3: The reviewer is confident but not certain that the evaluation is correct

**Scope:**

3: The work is somewhat relevant to the Web and to the track, and is of narrow interest to a sub-community

---

### Official Review · Reviewer_DPBn · 2023-11-24

**Novelty:** 5
**Technical Quality:** 6

**Review:**

# Summary

The paper proposes MeMPtec, a system for detecting malicious source code packages using metadata. The approach includes a feature extraction technique that identifies features that are easy to manipulate, and those that are difficult, and leverages them to find attack-resistant features. An experimental evaluation demonstrates significant reduction in FPs and FNs relative to state of the art feature selection approaches.

# Strengths

+ The paper identifies key properties that a feature needs to possess if it is to be used reliably for analysis without the fear of it being manipulated by an adversary.

+ The inclusion of features that are difficult to manipulate (DTM) leads to improved results over the state-of-the-art.

+ By constructing and testing with adversarial samples, the paper demonstrates how including the DTM features helps prevent misclassification (although it not completely eliminate the problem).

# Weaknesses

- The improvement over prior work in the base scenario (i.e., without adversarial/manipulated features) is not significant.

# Additional Comments

The paper makes an important contribution to the literature by identifying and leveraging features that are difficult to manipulate (based on well-reasoned properties) for the task of detecting malicious NPM packages. I was particularly surprised with how fragile existing work (and features) were in the face of adversarial samples, and how using the DTM features significantly improved performance in the same scenario.

That said, the values in Table 6 show that existing techniques are not far behind the proposed approach in non-adversarial situations. The paper attempts to explain this by describing how on absolute terms, the proposed approach generates fewer FPs than existing techniques (i.e., 2.6 vs 23). However, I don't see how the absolute number of false positives/negatives are even relevant, given that they become insignificant when considering the number of total samples. I don't think this diminishes the value of the paper, but rather that this unnecessary dissection of the FPs/FNs distracts from the overall message of the paper, and can be removed.

To summarize, this is a good paper that clearly outlines the properties that make features hard to manipulate, then systematically identifies features that exhibit the properties, and experimentally demonstrates improved detection capabilities even when the adversary manipulates metadata.

**Questions:**

Please clarify if I have misunderstood the intention behind discussing the number of FPs/FNs (when the percentages already demonstrated some improvement).

**Ethics Review Description:**

No ethical issues.

**Reviewer Confidence:**

3: The reviewer is confident but not certain that the evaluation is correct

**Scope:**

4: The work is relevant to the Web and to the track, and is of broad interest to the community

---

### Official Review · Reviewer_q17h · 2023-11-24

**Novelty:** 3
**Technical Quality:** 3

**Review:**

Summary:
The paper presents MeMPtec, a metadata-based model for detecting malicious packages in software repositories like NPM and PyPi. MeMPtec classifies metadata features into easy-to-manipulate (ETM) and difficult-to-manipulate (DTM) categories, improving detection effectiveness and showing resilience to adversarial attacks. Experimental results indicate a significant reduction in both false positives (up to 97.56%) and false negatives (up to 91.86%) compared to existing methods. MeMPtec addresses the critical challenge of securing software supply chains by leveraging metadata information.

Strengths:
+ Well-written and easy to follow
+ The metadata-related features considered are relatively comprehensive.

Weaknesses:
- Unavailable artifacts
- Missing some details
- Presentations need to be improved

**Questions:**

Most of the time I enjoy reading this paper. My biggest concern is that, in my opinion, the contribution of this paper is mainly to propose more features based on metadata information compared to other work. This makes the novelty and contribution of this paper less impressive.

The details about the used dataset are not clear. For example, how do you collect the dataset? Can you make sure that the labels (benign/malicious) of your data are all accurate?
By the way, the size of the dataset seems to be relatively small. It's not clear how well the MeMPtec performs in the real world scenarios.

Table 6 takes up a lot of space but is not well interpreted in the text.

It’s not clear what specific features are in the 𝐸𝑥𝑖𝑠𝑡𝑖𝑛𝑔_𝑡𝑒𝑐. It’s interesting to discuss that, which of your proposed features are more important compared to these features and thereby enhance the classification model.

There is no discussion of the limitations of the work.

Public release of the artifacts might help the readers who may want to extend this work further. Currently missing.

Presentations could be improved:

(1) Table 2 only lists the self-defined name of each metadata information, which is not clear to the readers to understand the specific information. It would be better to expand the table to three columns to present the metadata information, including self-defined names, descriptions, and examples.

(2) There are many self-defined terms in the paper. Sometimes it’s not easy to understand the term, e.g., what is 𝐷_𝑖𝑚𝑏_𝐿 in table 6? What is I𝑛𝑒𝑤 ?

(3) It is recommended that the figures be numbered in the order in which they appear in the text, however, figure 3 is mentioned before figure 2.

**Ethics Review Description:**

No Ethical concerns.

**Reviewer Confidence:**

3: The reviewer is confident but not certain that the evaluation is correct

**Scope:**

4: The work is relevant to the Web and to the track, and is of broad interest to the community

---

### Decision · Program_Chairs · 2024-01-22

**Decision:**

Accept

**Comment:**

## Summary
 This paper introduces MeMPtec, a metadata-based machine learning model for detecting malicious packages in software repositories like NPM and PyPi. The model classifies features into easy-to-manipulate (ETM) and difficult-to-manipulate (DTM) categories, significantly reducing false positives and negatives compared to existing methods. It represents a critical step in securing software supply chains, demonstrating how metadata can be leveraged to enhance package safety.

 ## Evaluation
 **Strengths:**
 1. **Clear Methodology:** The paper outlines a distinct approach to feature extraction based on metadata, improving the resilience of the detection model against adversarial attacks.
 2. **Notable Performance:** Experimental results show substantial improvements in detection accuracy over existing methods.

 **Weaknesses:**
 1. **Lack of Detail:** Some aspects, such as the dataset used and the specific features in existing techniques, need more clarification.
 2. **Presentation Issues:** The paper could benefit from clearer explanations of self-defined terms and improved figure organization.
 3. **Artifact Availability:** The absence of publicly available artifacts limits the ability for extended validation and replication.

 ## Suggestions for Improvement
 1. **Enhance Dataset Description:** Provide more details about the dataset's collection process and label accuracy.
 2. **Improve Presentation:** Clarify self-defined terms and align figures with their references in the text.
 3. **Release Artifacts:** Publicly share the code and data post-review to facilitate further research.

 ## Recommendations
 The paper presents notable advancements in the field of software security, particularly focusing on the detection of malicious packages through metadata analysis. The research is underpinned by robust methodologies, and its contributions are noteworthy. However, further refinements in terms of detail and clarity could significantly amplify its impact. To achieve the highest quality in the final paper, the Program Committee recommends appointing a shepherd to guide its finalization.

 ## Author's Rebuttal
 The authors have addressed several concerns raised by the reviewers, including clarifying their novel approach to feature extraction and the significance of their contribution in the context of adversarial attacks. They acknowledged the limitations related to the dataset and agreed to include more detailed discussions and clarifications in the final version of the paper. Additionally, they plan to make the code and data publicly available, which will contribute to the field's advancement by allowing for further research and validation.

 ---